# Economics (Only) Now: The Temporal Limitations of Economics as Part of a Critical Social Studies Pedagogy

**Neil Shanks**

Department of Curriculum and Instruction, School of Education, Baylor University, Waco, TX 76798, USA;
neil_shanks@baylor.edu

**Abstract:** This study speaks to the limited literature on economics pedagogy as part of a broader, critical, social studies pedagogy. Specifically, it seeks to understand the way that preservice teachers in an urban teaching program conceptualize the function of economics within social studies. Utilizing case study methods and a theoretical framework that intersects critical pedagogical tenets of social analysis with the idea of a counter-hegemonic stance, the study offers insight into the role of economics as part of a broadly critical social studies teacher education program. The results indicate that preservice teachers' purpose for teaching social studies and the function of economics were aligned in the mission to critically analyze society. However, preservice teachers' purposes for social studies extended beyond the function of economics into the past, and informed active citizenship for future action. These results show that economics can be a significant part of a social studies education practice that seeks to analyze society, understand the past, and take action for a better future. Unfortunately, limited familiarity and content knowledge inhibit a broader application of the function of economics. Social studies teacher education must purposefully integrate economics content into the exploration of the past and a discussion of future action for justice in order to combat prevailing content knowledge issues in preservice teachers and to help them reconcile their purpose for teaching social studies through economics.

**Keywords:** economics education; critical economics; preservice teachers; critical pedagogy; social studies; teacher education

---

## 1. Introduction

The landscape of social studies education literature is devoid of economics discussion in general [1]. History [2], and to a lesser extent other disciplines under the social studies umbrella, receive far more research attention than economics [3]. This is a problem given the potential for economics education to prepare students for social roles [4] or for the informed decision-making necessary for citizenship [1,5,6].

The existing literature on economics education points to two important themes, however, with respect to content knowledge and the function of economics. One, social studies teachers lack exposure to formal economics instruction; and two, the version of economics that is emphasized conforms to the dominant, neoclassical narrative [7].

While the literature is limited, nearly every exploration into teachers' content knowledge and previous experience with economics concludes that social studies teachers have limited coursework in economics. Scahill and Melican [8] surveyed AP economics instructors and found that "no more than 20 percent of respondents had received undergraduate instruction that many professional economists consider sufficient to teach AP economics" (p. 94) and nearly thirty percent of those surveyed had

taken three or fewer economics courses. The limited coursework for teachers of an advanced version of economics, ostensibly taught at a college level, heralds even more inadequate coursework for teachers of on-level economics. Data from New York teachers indicate that economics classes in general were taught by cross-disciplinary teachers who had taken, on average, 2.49 courses in economics and 13% of teachers had never taken an economics course [9]. Other studies have confirmed this relative lack of preparation via economics coursework [10,11]. This lack of coursework manifests in preservice teachers who are often unfamiliar with economics content [12,13], and are, in general, ill-prepared to teach the subject in comparison to other disciplines within social studies [14].

Additionally, the limited economics content that teachers are exposed to via coursework and teacher preparation is steeped in the neoclassical paradigm of economics. From the university classroom [15], to the textbooks in use in high schools and universities [16,17], to the national and state standards that frame the high school economics curriculum [18,19], neoclassical economics is the lens through which the world is read economically. This emphasis on neoclassical economics in coursework and the formal curriculum becomes the curriculum-in-use in economics classrooms [20], and thus the next generation of economics teachers not only has very limited content knowledge in economics, but the content knowledge that they have is almost exclusively confined to the neoclassical paradigm.

This study seeks to understand how content knowledge and previous experience with economics inform the way preservice teachers understand the function of economics within social studies education. Utilizing a theoretical framework that intersects the critical pedagogical tenet of *social analysis* with the idea of a *counter-hegemonic stance*, and as part of a broader case study of undergraduate preservice teachers, this study uses qualitative methods to explore this question in the specific context of an urban teacher preparation program.

*Theoretical Framework*

The preservice teachers in this case study reflected the contention by the literature that many preservice social studies teachers lack coursework in economics. However, they still had a number of important views on the function of economics within social studies education. Two themes were instrumental in understanding the way they viewed economics as a subset of social studies. Economics was important as a way to scrutinize society, part of a vision for social studies that conforms to critical pedagogy's emphasis on *social analysis*. This analysis occurs as part of a fundamental need to critique "texts, institutions, social relations, and ideologies as part of the script of official power" [21] (p. 4). This critique must start at the fundamental level where learners "are encouraged to question dominant epistemological, axiological, and political assumptions that are often taken for granted and often prop up the dominant social class" [22] (p. 8). Applying a critical lens to the world at this level requires one to read the world and the word dialectically [23], seeing the "world through the eyes of the dispossessed" [24] (p. 3). This involves critical analyses of race [25], gender [26], and capitalism [27] and the way these and other axes of oppression interrelate [28–32].

While the preservice teachers in this case study emphasized economics as a way to better analyze societal injustice, they often failed to include economics as a tool for understanding action in the present and future that would work against the dominant themes of the era [33]. In this way, the idea of a *counter-hegemonic stance* as a component of the ideological clarity they bring to their purpose for teaching was an important frame for interrogating their beliefs. A *counter-hegemonic* stance represents an answer to the fundamental question: Is the purpose of teaching to ensure the best possible outcomes in the current social order, or is it to revolutionize the existing order, in favor of a more democratic or emancipatory society? This question has a long history in American educational thought [34], and is vital to a thorough understanding of teacher purpose. Schooling has traditionally "functioned, in general, to transmit the dominant social order, preserving the status quo" [35] (p. 282). Teachers must decide whether they are to continue the transmission of this social order or if they are to transform it. There can be no more significant component to the teaching purpose than the choice of whether one will remain 'neutral on a moving train' [36]. The emphasis in this study was the extent to which and

the means by which preservice teachers explicitly advocated for and taught for transformation, or the way "they come to see the world not as a static reality, but as a reality in process" [23] (p. 83), that as teachers they must change. This *counter-hegemonic stance* is built through a teacher education program that emphasizes political clarity or "the ongoing process by which individuals achieve ever-deepening consciousness of the sociopolitical and economic realities that shape their lives and their capacity to transform such material and symbolic conditions" [37] (p. 98) as well as ideological clarity or "the process by which individuals struggle to identify and compare their own explanations for the existing socioeconomic and political hierarchy with the dominant society's" [37] (p. 98).

## 2. Methodology

### 2.1. Participants and Setting

This study made use of purposeful sampling in order to learn from "*information-rich* cases for study in depth" [38] (p. 230). In purposeful sampling, "[participants] are called in precisely because of their special experience and competence" [38] (p. 440). The questions addressed in this study inquire into the purpose of teachers who are participating in an urban teaching preparation program. Specifically, the questions deal with the purpose of economics, the purpose of social studies, and the purpose of teaching. The participants in this study were selected from a cohort of undergraduate students who entered an urban teaching program at a large, public university in an urban context. The Urban Teacher program was designed to prepare both undergraduate and master's teacher certification candidates in both English and social studies to appreciate the linguistic and cultural diversity of students in urban areas [39], drawing on their funds of knowledge [40] as the basis of instruction, and emphasizing critical multicultural citizenship [41]. The social studies coursework in particular emphasized the notion that there are dominant narratives [42] at work in the curriculum that maintain white, male, middle class norms, and promotes critical historical inquiry as a way to challenge those narratives [43]. Importantly, the program drew on a body of literature that challenges these narratives largely in history [44,45] or geography [46,47], but has a limited base of literature to draw on that questions the dominant narratives of economics within the social studies (e.g., the conception of 'man', scientific markets, absence of history, neoliberalism, etc.).

The participants in this study came from the undergraduate cohort within this program which does not constitute a major, rather students in the program complete the program's three semester course of study and required field experiences on top of a traditional academic major within any number of other schools or departments across the university. The program also has several unique characteristics which are relevant to this exploration.

The undergraduate program takes place over the course of a calendar year, beginning in the summer prior to an undergraduate's final year of collegiate coursework. The summer coursework includes two three-hour classes, literacy across the disciplines, and the sociocultural foundations of education. In addition to this coursework, preservice teachers enter into a field experience with Discovery, an Americorps program for future first-generation college students. The summer courses are designed to support the nascent understandings of students about critical literacy and the place of schools in society while also supporting them as they teach on a daily basis. The fall semester includes a social studies methods course and a 45-hour field placement in a local urban public school, followed by the spring semester which includes a three-hour teaching practicum that coincides with a full-time student teaching experience in a different local urban public school. In addition to their coursework and field experiences, preservice teachers are prepared for state certification exams through course content and extracurricular review sessions. The completion of this coursework and fieldwork, and the passing of the content and professional standardized exams are requirements for traditional teacher certification in the state.

Participant Descriptions and Backgrounds

Nora is a white female from outside of Chicago. She was a sociology major who always had teaching as a fall back plan, but eventually she "woke up. I keep saying this, I fall back on this, why do you think you keep coming back to this" (interview, 6/16/17) and realized that teaching was a way to incorporate her passion for sociology with a career, eventually developing an appreciation for teaching as a career in and of itself.

Tori is a black female from a major Texas metropolitan area. She brought up an eighth-grade teacher as a reason for getting into teaching, explaining that not only did he prepare students for standardized exams, but he was relaxed and helped students learn without feeling like it was an effort. She wanted to be a teacher like this, to enable students to achieve their goals. Particularly, her undergraduate education up to that point as a Youth and Community Studies major had pointed out the structural inadequacies of urban schools, which fostered a desire to help students in urban environments achieve their goals.

Ernesto identifies as a Mexican male, who was born in Mexico, but grew up in an urban area in Texas. His personal circumstances and living arrangements throughout his adolescence, helped him realize "the common thing that kept me out of trouble" (interview, 6/16/17) was school and a handful of good teachers who pointed out his potential. A film and digital media major, he sought to become a teacher like them for students such as him, and to avoid the negative perceptions that many have about urban schools and urban areas in general.

Lizeth is a Latina female who went to Head Start on the west coast, but lived the rest of her educational career in public schools in the Rio Grande Valley, including an Early College high school where she earned an associate's degree in biology while finishing high school. Shortly before matriculating to the state university, she decided that a career in science was "not going to make me happy in life" (interview, 6/22/17) and switched to a government major. This focus was generally the result of positive experiences with social studies teachers throughout middle and high school, both in terms of content, and their willingness to guide her through the college application process.

Jonny is a white female from a major Texas metropolitan area. She was a Youth and Community Studies major, whose parents were both in education including her father who was a social studies teacher. Her experience in high school as a tutor for standardized exams and as a co-teacher showed her that she "really enjoyed just being in the classroom" and her coursework in her degree pointed out the need for good teachers in "underfunded, under-resourced, diverse areas" (interview, 6/14/17).

Cristina is a Latina female who grew up near a major Texas metropolitan area, in an area she described as "the outskirts still the city but not downtown I guess it's a suburb", and was a first-generation immigrant and the child of immigrant parents. She was a social work major, but had a lifelong desire to be a teacher and so was pursuing her teacher certification in addition to her social work degree. Eventually, she wanted to do research on teaching and learning in urban areas that might culminate in opening up her own school that utilized the teaching styles and methods she developed while researching. Cristina was also extremely active on campus with a variety of student groups.

*2.2. Data Sources*

Participants were interviewed three times throughout the summer and fall. One interview took place in the early weeks of the Discovery teaching experience as preservice teachers were being trained for their roles in Discovery and began their coursework in the literacy across the disciplines class. This interview focused on background information about the participant including their demographic information, hometown, educational experiences, reasons for selecting the Urban Teacher program, their experience with economics academically, and their description of the purpose of social studies and the purpose of economics.

The second interview took place in the middle of the teaching experience at Discovery. At this point, preservice teachers had mostly completed their coursework in literacy across the disciplines and have taught for approximately three weeks in Discovery. This interview was designed to ascertain the

way their understanding of the purpose of social studies and economics had changed now that they had begun interacting with actual students. It also allowed for some member-checking of observation data from the first few weeks.

The final interview occurred after Discovery and summer coursework were completed. This interview included questions about the nature of critical pedagogy after a summer of teaching and two courses that made use of critical pedagogy both theoretically and in practice. It again asked about the purpose of economics and social studies, to determine whether the responses had changed as participants taught for six weeks and taken three teacher education courses. Finally, it allowed for extensive member-checking of observation data, and other codes and themes that had emerged from the data collected throughout the summer.

Discovery schedules numerous opportunities for professional development that were utilized to support participants and other students in the Urban Teacher program as they sought to implement a curriculum that aligns with their values and emerging understanding of the purpose of social studies and critical pedagogy. The first two weeks that preservice teachers were employed by Discovery consisted of a number of these sessions, designed to familiarize them with Discovery's procedures and prepare them for teaching. While the session topics were determined by Discovery, they were taught by Instructional Coaches who were doctoral students at the university, thus they offered another opportunity for preservice teachers to explore the practical application of their respective purposes for teaching.

Throughout the summer, preservice teachers participated in weekly professional development sessions conducted by the Instructional Coach. In these sessions, the preservice teachers generally planned their upcoming lessons with the help of myself, the Instructional Coach, and the other preservice teachers teaching the same content. These experiences contributed to understandings of the way that purpose, pedagogy, and content intersect in the collaboration and planning phases of teaching.

Preservice teachers also attended three different content knowledge professional development sessions, one focused on critical pedagogy, one on economics, and one on geography. Data from the first two content knowledge sessions were used in this study.

A variety of artifacts were collected in order to garner a richer picture of the data collected elsewhere, and to offer opportunities to conceptualize and reconceptualize themes. These included required reflections on their lessons for their university classes. They also included coursework from both literacy across the disciplines and sociocultural foundation classes in order to provide information on participant understandings of critical pedagogy and the purpose of social studies. Specific coursework artifacts included in-class products, written discussions, and assignments such as reflective journals and presentations. Of this coursework, their reflective journals in the form of blog postings were the largest source of data, particularly as they related to the purpose of teaching and the purpose of teaching social studies. Additionally, materials from professional development sessions such as writing and group products were collected, as were materials from other courses and classroom observations as they related to the emerging themes or fit within the conceptual framework.

### 2.3. Data Analysis

Freeman [44] asserts that all social science research "involves some sort of data identification, organization, selection, creation, recognition and some sort of transformation of what is identified, organized, selected, created, recognized into a statement about the topic of inquiry or 'findings'" (p. 3). Therefore, analysis is the process that guides this identification and transformation. As opposed to quantitative research, where much of the analysis occurs at the end of the inquiry, "data analysis in qualitative studies is an ongoing process" [48] (p. 437) and thus data are reviewed and reflected on throughout the collection process, and should therefore alter or enhance the data collection process as well [49]. It is also important to consider the political dimensions of analysis. By choosing a mode of analysis, a researcher is inherently endorsing an epistemology and ontology [50] and thus endorses a

perspective consciously or unconsciously. By specifying an analytical framework and linking to the underlying epistemology, we can begin to "intentionally disrupt the 'qualitative positivism'" [50] (p. 5) that is common in many qualitative reports.

Rather than identify these frameworks as analytical methodologies, Freeman [44] argues that we should think of them as ways of thinking, which allows researchers to avoid the reductionism that can be involved in limiting oneself to a singular analytical framework. While this study made use of two main ways of thinking, I describe them "not for the purpose of fixing them as methods, but to enable their circulation, adaptation, and even, their transformation" (p. 5). In particular, I drew on thematic or categorical ways of thinking as well as narrative ways of thinking. Thematic/categorical thinking focuses on "searching through the data for themes and patterns" [51] (p. 187) which "helps to separate out units of data that can stand alone often as a way to contrast or relate them to other units of data" [50] (p. 8). Narrative ways of thinking vary across disciplines, but focus on "how the storyteller links experiences and circumstances together to make meaning" [51] (p. 186). For the purpose of this study, I attended to the way preservice teachers engaged with and described narratives in social studies and society more broadly. This way of thinking utilizes literary devices such as "topics, plots, themes, beginnings, middles, ends, and other border features that are assumed to be the defining characteristics of stories" [52] (p. 226). By thinking about the stories they told (and did not tell, or did not position as stories), I could better analyze their conceptualization of the purpose of economics and its relationship to critical pedagogy.

Initial data were explored by "reading and thinking and making notes" [48] (p. 438) or memoing which can take a variety of forms as the researcher attempts to parse meaning and select data for coding [51]. Early coding was rudimentary and strived to be simple "knowing that with use, [codes] will become appropriately complex" [47] (p. 191). As more data were analyzed and more sources of data were coded, codes were categorized and organized as necessary.

## 2.4. Researcher Positionality and Limitations

In case study research, "[t]he researcher is the primary instrument of data collection and analysis. This has its advantages" [49] (p. 52) including a particular expertise or familiarity with the unit of analysis. In my case, my interest in economics and urban teaching stems from my time teaching economics in an urban setting and my experience with marginalized students beginning to understand structural inequality and speak back to their conditions via the language of power. Likewise, my interest in preservice teachers stems from my desire to become a teacher educator and my time working as an instructor and field supervisor of preservice teachers. These strengths of expertise and familiarity however, can be a double-edged sword as my biases can affect the way I interpret responses and my position as authority figure in the program could color the responses of participants. In addition, I conform to a host of societal norms that have and continue to be used to 'other' marginalized groups. Any white, male, cis-hetero researcher should be concerned with persistent issues of colonialism in research and questions of who speaks through and who benefits from their research.

Perhaps the most significant component of my positionality, with respect to the conditions of power in qualitative research was my role within the Urban Teaching program. In relation to these preservice teachers, I either was or would serve as a Teacher's Assistant, course instructor (though this did not occur until the following Spring), field supervisor/evaluator in the Fall semester, mentor, job reference, state exam tutor, and confidant. While I cannot expect to know every way that my position affected the data I collected, I include the following examples of how my influence and relative authority over these preservice teachers might skew the data I collected. For convenience, I used the class Learner Management System to send out a voluntary request for interviews. I conducted professional development sessions on economics during Discovery department meetings, including choosing a reading to assign. I conducted a methods class on economics during their Fall semester methods course which is typical of my service as a Teacher's Assistant, but relevant to this discussion

nonetheless. I often discussed their purpose for teaching and teaching social studies as their field supervisor in an attempt to promote reflective thinking and develop them as pedagogues. While I respect the autonomy and agency of these preservice teachers and believe I did everything in my power to foster a relationship where they felt comfortable to participate or not participate in the study and to be as honest as possible in their responses, I cannot assume that my relative power in the Urban Teaching program was without impact on our shared data.

Additionally, a small sample size of this case and its specific context weigh heavily on the generalizability of the findings. While generalizability is "a term that holds little meaning for most qualitative researchers" [53] (p. 102), the preceding details of the context and participants must be kept in mind while considering the utility of the resulting discussion.

## 3. Results

This study is framed around the research question that asks how content knowledge and previous experience with economics inform the way preservice teachers understand the function of economics within social studies education. The data revealed that preservice teachers had limited experience with economics and limited content knowledge which helps to understand two important themes, summarized in Figure 1. First, preservice teachers in this study articulated an expansive conception of the purpose of social studies, including the need to analyze society in the present, critically evaluate the past, and promote active citizenship. Second, economics was an important component of social studies as a means to analyze society, however, this analysis was largely confined to the present. They rarely analyzed the past through an economic lens despite their stated intention to use social studies to do just that. Economics was also not included in their ideas about social studies as an active practice that sought to alter the future of society for justice. While their purpose for teaching social studies often centered on a count-hegemonic curriculum; economics was rarely included in this vision of social studies.

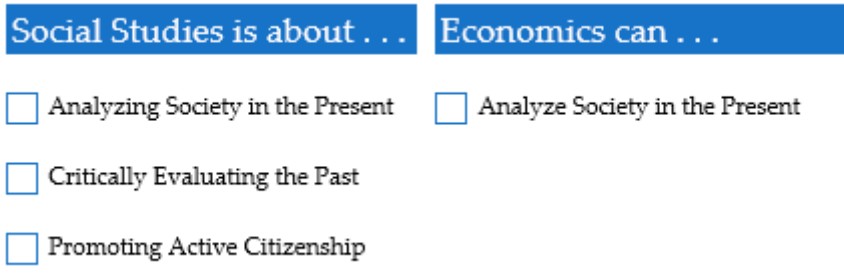

**Figure 1.** The Function of Economics within Social Studies Education.

### 3.1. Content Knowledge and Previous Experience with Economics

The participants in this study had very little experience with economics, reflecting the findings of a number of studies that explore the economics backgrounds of social studies teachers [12–14,54–58]. Four of the participants had taken a single semester course in economics in high school, and two had not taken an economics course at all. Of those that did take a semester in high school, Tori and Ernesto took Advanced Placement macroeconomics, while Cristina and Jonny took an 'on-level' course that complied with state social studies standards. Tori and Lizeth did not take any economics classes either in high school or college. Tori's private school did not require the class as part of their curriculum, and Lizeth's early college high school focused on classes needed to earn an Associate's degree upon graduation which limited the number of social studies classes taken in her degree plan.

Given this lack of exposure to economics via formal curriculum, it is no surprise that preservice teachers ranked their content knowledge in economics toward the bottom when compared to other disciplines within social studies. For Tori, economics would rank "at the bottom" (interview, 6/16/17) of disciplines such as history, geography, and political science. She explained that she did not "remember much, because I didn't pay attention much, because I didn't care about it as much"

(interview 6/16/17). Jonny likewise expressed her lack of content knowledge as "pretty low" because it did not "click in my head as well as other subjects within social studies do" (interview 6/14/17). Cristina and Ernesto both ranked their content knowledge in economics at the bottom of the disciplines within social studies. Lizeth, expressed her displeasure with economics (despite not having taken a class in economics) by saying it was her least favorite, and it was the subject she had the least knowledge in. Nora was the only preservice teacher to qualify her ranking, and described her content knowledge as:

> Pretty far down there. My understanding of economics is very much from a theoretical perspective, talking about Marx or functionalism and that kind of stuff, that's where I get my understanding of economics. It would definitely be below history and government. (interview, 6/16/17)

This understanding of economic theory outside of the traditional, neoclassical economics environment stemmed from her high school history classes, and undergraduate sociology courses as part of her major.

In a professional development session during their summer field experience, five of the six preservice teachers were exposed to the Voluntary National Content Standards in Economics (VNCE) [19]. These standards adhere closely to the neoclassical paradigm, are used in many states as a foundation for individual state standards, and include 20 core concepts such as scarcity, decision making, trade, income, and fiscal and monetary policy. They were asked to read the summary of each standard in the table of contents, then mark whether they were familiar with the content, would be comfortable teaching the content or both. Of the five who attended, Tori, Jonny, and Nora had familiarity with nine or more of the twenty standards, and were comfortable teaching six of them on average. Cristina and Lizeth were comfortable with fewer than nine standards each and did not feel comfortable teaching any of the standards. The standards that preservice teachers were most familiar with were scarcity, specialization, and entrepreneurship, with each garnering a response from four out of five preservice teachers. The VNCE intentionally represent a neoclassical vision of economics [18,19], and these three standards adhere to that paradigm by emphasizing the economic way of thinking [59] and capitalist notions of entrepreneurship as a factor of production. No single standard had more than two preservice teachers indicate they would be comfortable teaching it.

### 3.2. Articulations of the Purpose of Social Studies

3.2.1. Social Studies Now: Awareness, Understanding, and Conceptualizations of Justice

Throughout this study, preservice teachers articulated a conceptualization of social studies that was used for *social analysis. Social analysis* includes the ability to see and question forces that maintain dominance and oppression in society [22], and for these preservice teachers this component of critical pedagogy manifested in three ways. They felt social studies could would make students aware of their society, understand the way that it functioned, and interrogate it to determine whether or not it was functioning in a just manner. At times, they described this function as a specific component of critical pedagogy, and other times it they simply alluded to it, but these three elements of *social analysis* were important components of a critical pedagogical curriculum "shaped by problems that face teachers and students in their effort to live just and ethical lives" [60] (p. 17). These elements of *social analysis* were present in conversations about of the purpose of social studies and economics in interviews and professional development sessions, and were also present in preservice-teacher generated class artifacts and blogs.

There was a consistent emphasis on social studies as a tool to build awareness. When asked about the purpose of social studies, Nora responded that "I'm really just trying to study society. I thought from social studies, you get to do that" (interview, 6/16/17) in a way that shows students "why certain things happen the way they happen, [and] how society operates" (interview, 3/16/18). Ernesto likewise responded that "it is the study of what's going on in the world" (interview, 6/16/17). In the

first of three professional development sessions, which focused on the role of critical pedagogy in social studies and economics, students responded to quotes from an article that described an instructional plan about teaching wealth distribution via critical pedagogy [61]. Some of their responses included their purpose for teaching social studies as a way to make sure students are informed about "words like racism, ageism, oppression, etc." because "I want [students] to be well informed" (PD1 artifact, 6/22/17). Additionally, they saw social studies as a way to show "contemporary issues which they can write and discuss their thoughts on. I want [students] to think about the world and time they live in" (PD1 artifact, 6/22/17). In blog postings, Lizeth wrote that she wanted students to "be able to think critically about current events" (blog posting, 6/1/17). These descriptions of social studies for *social analysis* revealed the idea that social studies was a lens to view the world, and a way to begin to explore themes of the current era.

While the preservice teachers had a great deal to say about the way that social studies could be used to be aware of society and social issues, they quickly moved on to describing how social studies could be used to determine the way that forces in society worked, and did so in interviews, class discussions, and blog postings. Ernesto emphasized this while decrying his peers' "lack of understanding of the system. It's a lack of understanding of how everything around us works and how society works as well" (interview, 6/16/17). Nora echoed this functionalist imperative by saying, "a big part of [the purpose of social studies] is to try to understand how your society works and also projecting that into a more global view" (interview, 6/16/17). In Tori's words, social studies was an important discipline because "I want to know all about the system, the situations and how to handle them not just education, but more current situations that we encounter" (interview, 6/16/17), evincing a desire to understand the impact of social forces specifically in the present. For Lizeth, the breadth of social studies allowed students to see how social forces operate, she stated that social studies was "so broad and extensive there's so many different things you get to learn within this topic" and therefore it allowed students to "learn and analyze all these different types of context and events" (interview, 6/22/17). In a class session, students generated a list of reasons for teaching social studies that included its utility in understanding "policy", "conflict", "movements" and "why society acts the way it does" (class artifact, 6/13/17).

In blog postings, preservice teachers further developed these ideas. Ernesto talked about social studies bringing about a "newfound sociopolitical stance since he/she will have thought about their unique place in the world and the various conflicts that stem from it" (blog posting, 6/1/17). Similarly, Jonny felt social studies teachers should "strive to teach socially conscious curriculum that empowers students to be thinkers" (blog posting, 6/7/17). Social studies, then, afforded these preservice teachers the opportunity to not only analyze society as it exists, but to conceptualize the methods by which it functions. This *social analysis* would be fundamental to the idea of justice that preservice teachers would later articulate, a necessary theoretical underpinning to their *counter-hegemonic stance*.

Perhaps the most prevalent descriptor of the purpose of social studies was a way to analyze society within a framework of justice. Preservice teachers talked about social studies as a way to critique injustices that they saw in the world, or to promote a vision of social justice that would later inform the transformative purpose of social studies through their *counter-hegemonic stance*. These descriptions of the purpose of social studies as a way to shine a light on injustice were demonstrated in blog postings, interviews, and professional development sessions. Importantly, their conceptualization of justice followed at least two paths: justice through student voice and multiple perspectives, or recognizing the value of student experience and including marginalized groups in the curriculum; and justice through *social analysis*, or the ability to "perceive critically the themes of their time" [33] (p. 6).

In a blog entry on the importance of discussion in social studies, Tori described social studies as a place to "remember that somebodies voice is going to be left behind or not fully explored. Having these authentic discussions will allow for the underrepresented to have a spokesperson of sort" (blog posting, 6/15/17). Nora emphasized this vision of justice as well, writing, "we must strive to teach with cultural relevance in mind, to teach with the understanding that [social studies] is not a single story and that

our students have the right to be represented" (blog posting, 6/12/17). After having taught for several weeks, Cristina reiterated her purpose for teaching social studies by saying "I still feel so very strongly about [social studies] being about helping students to self-advocate, and to promote social justice" (interview, 7/13/17).

Justice also included notions of including marginalized perspectives in the curriculum. Nora documented her vision of social studies as a way to "consider whose voices are not heard, and whose faces are not shown" and to combat the "current social studies curriculum [that is] dominated by the narratives of one person – the exceptional white man" (blog posting, 6/12/17). Ernesto described the purpose for teaching social studies as a way for teachers to be more culturally relevant because "[c]ulturally relevant teachers will be more aware of their perceptions on their own students and view the students' diversity and individual differences as an opportunity rather than a worry" (blog posting, 6/2/17). Likewise, Jonny wrote that "[a]s an aspiring Social Studies teacher, it is my goal to make history as real and well-rounded as possible" (blog posting, 6/12/17). By emphasizing justice through inclusiveness in social studies, and emphasizing cultural relevance to students' lives and curriculum, preservice teachers voiced a purpose for social studies that extended beyond the official knowledge [62] and represented a burgeoning ideological clarity [37] about the nature of marginalized students and traditional representations in the curriculum that would inform their *counter-hegemonic* stance.

Quoting and responding to an article on critical pedagogy in a professional development session, one of the preservice teachers wrote, "This is what and who I want my students to be: 'critical, self-reflective, knowledgeable, and willing to make moral judgments and act in a socially responsible way' [61] (p. 238). So, modeling and teaching that to the best of my ability is really important in my classroom" (PD 1 artifact, 6/22/17). In a blog entry, Jonny outlined her goal for social studies by saying that "teachers should strive to teach socially conscious curriculum that empowers students to be thinkers" (blog posting, 6/7/17). Lizeth similarly described a purpose for teaching social studies as ensuring "that my kids are interested in what we are learning but also for them to be able to think critically about current events" (blog posting, 6/1/17). Jonny explained in an interview after she began teaching that her students have started to attain this purpose for social studies and that:

> a lot of them, in their papers wrote, "I used to just believe whatever my parents said, I used to just believe whatever my friends said, but now that I'm looking at some of the stuff that we're doing in class, like I know that – I just took something at face value instead of being critical of what I'm seeing". That was really awesome. (interview, 7/11/17)

For these preservice teachers, justice within their purpose for social studies meant a focused and direct use of the discipline to critically evaluate the world around them.

### 3.2.2. Social Studies Then: Critical Evaluations of the Past

Given the traditional emphasis on history as the preeminent discipline within social studies [2,63], it is not surprising that preservice teachers were quick to extend the purpose of social studies beyond the present as a way to understand and learn from the past. In keeping with the critical orientation of the teacher preparation program that these preservice teachers were a part of, they insisted on a vision of social studies that critically evaluated the past and challenged dominant narratives [42,43] that have been promoted through a nationalist, Eurocentric curriculum [44,64,65].

While a number of preservice teachers responded to questions about the purpose of social studies with some variance of the "whole cliché, like, you have to know what the past looks like in order to prepare for the future" (Jonny, interview, 6/14/17) or to "learn about the past or the history of our country or the people before us, learn from their mistakes and from those mistakes, try to make a better future" (Cristina, interview, 6/10/17); their responses quickly fashioned this understanding into a way of critiquing the past and present. After describing the purpose of social studies as "a real way to talk about things from the past" (Lizeth, interview, 7/30/17), Lizeth went on to give an example about a conversation with her boyfriend about what seemed like an increase in protest marches. She said,

"pretty much everything that has changed has come to the point of people having to [march] and protesting to get something to be changed within the law" (interview, 7/30/17). This quick turn from simply knowing about the past to avoid the "mistakes of the past" (class artifact, 6/13/17) to understanding action in the past in pursuit of justice would be emphasized in blog postings and professional development sessions as well. Often this context and analysis involved "understanding the Latinx civil rights movement" (PD1 artifact, 6/13/17) or other protest movements in contrast to "most social studies classes [which] focus too much on dates/people" (PD1 artifact, 6/13/17). Lizeth concurred, writing that "[h]istory should be about thinking critically and analyzing the situations, people, objects and ideas in the past" (blog posting, 6/6/17). Nora elaborated on a critical use of social studies to evaluate the past by writing that "[i]t is important to recognize the significance of the social contexts in which historical events occurred, and not solely the people (or more specifically, the men) we have thus far considered to be significant" (blog posting, 6/1/17). Following up on this point, Nora reiterated that she saw "our understanding of social studies now as very limited we kind of take away the greater context by just focusing on the [state standards] or just focusing on specific people" (interview, 3/16/18).

In addition to an emphasis on the purpose of social studies as a way to explore the past in its complexity, there was a specific emphasis on the concept of studying the past in order to contest the "Euro-centric and white American experience [that] is dominant in most American classrooms" (Ernesto, blog posting, 6/1/17). For these teachers, it was "imperative to consider who is absent from any given narrative. It is important to consider whose voices are not heard, and whose faces are not shown" (Nora, blog posting, 6/12/17). Jonny reflected on her own experience in school and on reading the introduction to Takaki's [65] *A Different Mirror*, recalling, "[my teacher] would always ask the class who was missing from or misrepresented in the source. He bluntly told the class that, 90% of the time, the answer would be as simple as women, people of color, or children" (blog posting, 6/12/17). Lizeth challenged dominant narratives through classroom practices that are "monologic and only talk about one perspective", which corresponded with "how I was taught to think until an eighth grade social studies teacher came along. He challenged us to question everything that had been taught before and why we listened to it without questioning it just because a teacher told us that is what happened in history" (blog posting, 6/15/17). These emphases on the purpose of social studies as a means to challenge traditional ways of teaching history and dominant narratives within the curriculum were often a pretense to engage in the next theme that arose from the data: social studies for active citizenship.

### 3.2.3. Social Studies in the Future: Active Citizenship for Change

Preservice teachers in this study conceptualized the purpose of social studies as both a component of good citizenship, and a springboard to action in the face of injustice. Social studies as a discipline, and public education in general, have often been described as vital components of preparing an active, knowledgeable citizenry [66–68]. This imperative leads to questions about the kind of citizen that schools prepare [69], including personally responsible citizens, participatory citizens, and justice-oriented citizens. Abowitz and Harnish [70] categorize citizenship frameworks as falling into more common categories of civic republican or liberal frameworks and less common transnational and critical citizenship discourses. Important to this study is the category of critical citizenship discourses which "raise issues of membership, identity, and engagement in creative, productive ways" [70] (p. 666). This engagement is in keeping with ideas of critical multicultural citizenship, where "citizens engage in meaningful deliberation about the ideals of democracy and gaps in its realization in everyday life and pursue social action to close these gaps" [41] (pp. 222–223). In keeping with these elements, an early literacy class session produced a brainstormed list of the purposes of social studies, and among the first items listed were "Active, democratic participation" and "civil disobedience" (class artifact, 6/13/17). The idea of social studies for citizenship and social studies for action were prevalent in blog postings, interviews, and professional development sessions.

Citizenship was often explicitly stated as a purpose for social studies, and usually came up immediately when asked what social studies was for. Cristina described the purpose of social studies as helping "to become better citizens" (interview, 6/10/17) who "are able to make better, well-informed decisions when it comes to politics and to give back to their community" (interview, 3/15/18). In a blog post, Jonny wrote that dialogic teaching fit her desire to "promote democratic participation [and] help students build their skills as a productive citizen in a democratic society" (blog posting, 6/15/17) Later, she continued to expand on her purpose for teaching social studies by saying "I strive to empower my students to be active citizens and practice self-advocacy through the use of problem-posing and liberating education" (blog posting, 7/13/17). Lizeth similarly wrote about "one of my biggest and main goals will be to get my students to become active participants in our country's democracy I like knowing that I am getting prepared to get students to think critically and be active citizens" (blog posting, 6/28/17). Also, in a professional development session, preservice teachers utilized this language to explain that one of their "main goals as a social studies teacher" was to get "students to be active citizens through democratic participation" (PD1 artifact, 6/13/17). These references to citizenship seem to fall under the category of participatory citizenship, where "good citizens [are] those who actively participate in the civic affairs and the social life of the community at local, state, and national levels" [69] (p. 241), yet based on their understanding of the use of social studies to be aware of social forces, to understand how they operate, and to promote justice there is more to these descriptions of active participation in democracy. This is alluded to when Jonny begins to reveal a burgeoning counter-hegemonic stance [37], by beginning to cite specific elements that have informed her ideological clarity, including her desire to pursue problem-posing [23] and liberating education [71]. Their deliberation over the ideals of democracy that were previously explored and their desire to take action based on these ideals reveal a more critical citizenship [70,72]. These elements of a critical and active vision of social studies became clearer when preservice teachers moved beyond the explicit language of citizenship and spoke and wrote of what impact they wanted social studies to have on the future of society.

Many of the visions of the future relied on a purpose of social studies that took the lessons of the past and applied them to future action. For Cristina, "the purpose of social studies is to learn about the past or the history of our country or the people before us, learn from their mistakes and from those mistakes, try to make a better future for future generations" (interview, 6/10/17). Nora built off of the contention that social studies was about understanding "this is what happened, this is why it happened. Then maybe trying to project that into the future what we can do better" (interview, 6/16/17). Ernesto followed the same general path, stating that social studies was important "for a student not just to learn from what mistakes we made in history but also to understand what we can do now in order to develop our future" (interview, 6/16/17). Influencing the future was a significant theme for Lizeth who saw social studies as a way "to try and get to that point of change" (interview, 6/22/17) and "to try and change something right now so that it can affect people differently in the future" (interview, 7/30/17). Beginning to place their temporal focus for social studies in the future lead preservice teachers to take a stand based on what they wanted social studies to do for the future. In blog postings, preservice teachers expanded this active notion of social studies into overt declarations of their political and ideological clarity.

Tori wrote about the importance of understanding that "there is no way a teacher can be neutral in their political stance" and that social studies therefore should "allow for subordinate groups to participate and be aware but also to step outside of the personal into the sociopolitical" (blog posting, 6/7/17). Nora also described teaching as "a political action" and emphasized the need for social studies teachers to "speak out against injustice, both in and out of the classroom" (blog posting, 6/28/17). Ernesto included an example of this from his planning prior to teaching. After describing social studies teaching as "guiding our students rather than giving them what to do" he wrote about his desire to have "students develop their own protest in my class. As their educator I will supervise and guide, but the students are responsible for choosing which issue they'd like to pursue and what

form of protest is appropriate for it" (blog posting, 6/15/17). These responses show a clear articulation of a social studies purpose founded on social analysis that seeks to use classroom practice to achieve a nascent form of praxis, or action on the world to combat injustice. The same, unfortunately could not be said with regard to their articulation of the function of economics within social studies.

### 3.3. Articulations of the Purpose of Economics

#### 3.3.1. Economics Now: Awareness, Understanding, and Conceptualizations of Justice

In a reflection of their purpose for teaching social studies, preservice teachers described economics as a way to better make sense of the present. Specifically, they emphasized the unique role that economics plays in *social analysis*, as a component of social studies that performed the same role. The way that they articulated *social analysis* as part of the purpose of economics aligned with their expressed purpose of teaching social studies in that it could build awareness, increase understanding of social forces, and allow students to better conceptualize justice. These articulations were present in interviews, professional development sessions, and blogs.

Economics education afforded unique tools that could aid in an awareness of society. Tori described the purpose of economics as allowing students to understand the way the economy works in such a way that "a student could actually go out and explain it to someone else" and "to make it more relatable" (interview, 6/16/17). For Nora, economics "ties into understanding of how society works, I think economics is fundamental part of that" (interview, 6/16/17). Cristina felt that the purpose was to "know about the system or even what are the rights and rules about it" (interview, 6/10/17). For Ernesto, the purpose of economics was to "be able to understand this complex system that we have that pretty much involves trade and income and everything that allows us to have a more sustainable life and a livable life" (interview, 6/16/17). In the first professional development session, preservice teachers were prompted to think about how critical pedagogy and economics fit together. Their responses included a number of references to economics as a way to build awareness of social forces such as "knowing how economic systems work," "understand wealth distribution in the real work", and "providing the tools needed to begin understanding the 'power' and the 'oppression'" (PD1 artifact, 6/22/17). These responses indicate that economics was conceived of as a way to further understand society, and to do so with the specific attention to economic themes.

The emphasis on social studies as a tool to enhance consciousness of the machinations of social processes were evident in descriptions of the purpose of economics as well. In interviews and professional development sessions, economics was continually described as a path to understanding the function of social processes. In a written response to an article used in the first professional development session, preservice teachers described economics as a way to "teach students the fundamentals of knowing how economics systems work [and] how the systems impact their communities" (PD1 artifact, 6/22/17). In interviews, preservice teachers used similar descriptions. Nora thought about the purpose of economics in terms of understanding "capitalist society and how it actually works in practice" (interview, 6/16/17). Lizeth also found economics to be of use in understanding the function of society, particularly with the minimum wage and inflation. For her, economics helped understand:

> How all these things just keep increasing in price yet the minimum wage doesn't have that much of a difference over time, and how that affects so many people right now, so many people who are just struggling to get by and this minimum wage just isn't helping. (interview, 7/30/17)

Cristina sounded a similar conspiratorial note by stating that economics could be a tool to investigate problems with the way the system functioned, saying "I always feel the system is up to something and I personally feel that it's to keep people ignorant you're messing around with their money, but you don't inform them about how does the money work or how does the system work" (interview, 6/10/17). These responses show that the preservice teachers were not only using economics

to see the world as it is, but to consider the processes that keep it that way. This understanding of systemic functions also informed their vision for economics as a way to think about the relative justice in society.

Descriptions of the purpose of economics as a means to conceptualize justice were also common, in keeping with their purposes for teaching social studies. Preservice teachers continually described the purpose of economics as a lens through which justice could be analyzed with respect to class, income and wealth. Economic justice is a sub-concept of social justice [73], which includes recognizing intergroup economic disparities and considering redistributive measures to address these disparities [74], and fulfills tenets of critical pedagogy that are "fundamentally concerned with understanding the relationship between power and knowledge" [22] (p. 144). These components of economics for justice were prevalent in broad terms as well as with respect to specific issues analyzed through economics. Nora included broad ideas of systemic justice, recognition, redistribution, and power as she described the purpose of economics as:

> part of being cognizant of systems at work that dictate the opportunities they have as students or things available to them. Just making sure that they're aware of those so that if they see some kind of injustice, again, it's like the social justice mindset, you see an injustice maybe you can do something about it because you understand how that injustice came about, from an economics perspective. (interview, 6/16/17)

Jonny also felt that economics was a way to touch on "classes and what it means to be in a certain class and the implications that has in social life" (interview, 7/11/17), demonstrating that economics could analyze the way that economic class had the power to affect other aspects of life. In the first professional development session, preservice teachers explained that economics should "produce those self-reflective and knowledgeable citizens" who are "[c]ritical of economic policy and procedures, [and] sympathetic to economic issues" (PD1 artifact, 6/22/17). It should also allow students to explore "how economic systems work, [and] also discuss and question how the systems impact their communities" (PD1 artifact, 6/22/17). These responses show an emphasis on economics as a way to build the critical, analytic perspectives to address local and relevant injustices.

There were also specific topics that economics could be used to explore. Cristina saw economics as a tool to analyze how "some schools get more money than other schools" yet this is unquestioned because people "don't have much knowledge about economics or how does the money flow or the politics behind it" (interview, 6/10/17). Lizeth considered economics as not limited to "business and finance processes", but offering a way to explore "a lot of economic disadvantage topics in there, like maybe even some types of urban development issues that are better taught through economics" (interview, 6/22/17), again demonstrating the use of economics to build recognition of injustice, with particular attention to urban development. Ernesto, in an interview reflecting on his experience teaching, talked about how economics is more than just simple dollars and cents, but allows for consideration of broader themes:

> I was once talking to a student about [economics] and she kind of brought up the idea how one thing she's always questioned is why is it that when we talk about immigrants or disenfranchisement in our society, we always seem to kind of think on the economic benefits that we gain from them, rather than just perceiving them as human beings, as people who can become part of our culture and society but instead we try to develop ideas like "how can we sell it to the public, to be beneficial to us" I think it's valuable, in the sense that, then you can ask these big idea questions about today. (interview, 7/21/17)

This was a trenchant analysis of the inherent value that policy makers and politicians place on economics. He is arguing for a more human approach to policy, rather than reducing immigrants to dollars and sense as debate rages over this particular political issue.

Economics functioned within this vision of social studies by a lens to see society, a framework for understanding its operation, and specific areas to emphasize when conceptualizing justice in society.

Despite limited content knowledge, economics was still an important component of these preservice teachers' purpose for teaching social studies when applied to the present time.

3.3.2. Economics (Only) Now: The Conspicuous Absence of Economics in the Past and Future

In contrast to the alignment between the way that preservice teachers described the purpose of both social studies and economics in the present, a temporal shift to either the past or the future caused a disjuncture between the purposes of social studies and economics. While the purpose of social studies was fluidly articulated as having relevance in the past and to inform action in the future via citizenship practices; discussion of the purpose of economics almost never veered from the present described in the previous section. This failure to conceptualize the utility of economics in the past is in keeping with a general failure of the discipline of economics to explore the past, either in economic models (c.f. [75–77]) or the history of economic thought [15,78–80], but the noticeable gap between social studies informing civic action and the silence surrounding economics' potential role in that action is significant. Freire writes that in order to achieve praxis, people must "emerge from time, discover temporality, and free themselves from 'today'" [33] (p. 4). Thus, the rupture between social studies and the function of economics in the past and future will inhibit the ability of preservice teachers to turn their *counter-hegemonic stance* into "some type of action to 'subvert the system' and do right by their students" [37] (p. 118). The juxtaposition of the purpose of social studies and economics in the past and future was evident in interviews about of the purpose of social studies and economics, professional development sessions, as well as preservice-teacher generated class artifacts and blogs.

The purpose of economics was rarely described in terms of understanding either the past or fomenting active citizens ready to create a better future. There was only one mention of economics for this reason in interviews and a handful more in a professional development session. Later, some preservice teachers commented on the schism between their purpose for social studies and the function of economics within this purpose when specifically prompted to in interviews. Nora was the only preservice teacher to talk about the purpose of economics as having some utility for active citizenship, describing her purpose for economics as part of her main goal "to help students realize their full potential. Whether it's strictly economic or whether it's the type of participation they're going to have as a member of society, whether just democratic or just a human" (interview, 6/16/17). This quote demonstrates the only response to a direct question about the purpose of economics that indicated a belief in the function of economics as a part of democratic citizenship, despite the a relatively extensive body of literature on the subject [5,6,81,82]. Importantly, it took the introduction of an article that explored the intersection of critical pedagogy and economics [61] to prompt a range of responses that began to talk about the purpose of economics in terms of understanding the past, and as informing an active citizenship. In this setting, preservice teachers wrote that:

> Economics is political—it is wrong to say that exercising your rights as a citizen (or even your role as citizen) does not influence economics directly/indirectly. A well informed student (citizen) will be able to realize/influence economics within their community. (PD1 artifact, 6/13/17)

This shows that when exposed to ideas about the juncture of economics and action, they began to consider the way that citizenship and economics might be linked, something that had not come up before. They also interpreted a quote about the importance of teaching wealth distribution by saying it is "important to teach students not just data and terms, but teach them in a way that they can understand wealth distribution in the real world. That will help them understand those problems and maybe even help solve them" (PD1 artifact, 6/13/17). This idea of economics as a way to inform action was continued in a response to another quote about Henry Giroux's vision of critical pedagogy:

> [i]n relation to economics, teaching students to be critical thinkers, be sympathetic, and intervene with major problems will help students better grasp econ as a subject.

Critical of economic policy/procedure, sympathetic to economic issues, and activist to fix economic problems. (PD1 artifact, 6/13/17)

When given the space and material to reflect on the potential for economics to perform a more critical function within their purposes for teaching social studies, the preservice teachers were far more likely to conceptualize a discipline of economics that analyzed society with an intent to reshape it.

In order to address the emerging understanding that these preservice teachers were struggling to match their purpose for teaching social studies with economics, a professional development session was conducted to address this absence. The session was a manifestation of the need to address content knowledge in economics, while simultaneously expanding the perceived utility of economics into the past and the present. Briefly, the session used physical markings in a hallway as a bar graph. Students representing income quintiles illustrated current income inequality. Later, students representing racial wage gaps in 1963 moved to current racial wage gaps to illustrate the need for continuing the economic justice themes of the March on Washington. Finally, students represented wages of agricultural workers at the time of the Delano Grape Strike, their demands, and current minimum wage standards. The goal of this session was to address the economic concepts of inflation and income while using economics to tangibly illustrate injustice and inform future action.

In interviews, preservice teachers reflected that this session helped them integrate unfamiliar content in ways that fit their ideal social studies practices which critically evaluated the past and informed active citizenship. Cristina described her struggle with economics by saying "I'm so insecure to teach [because] I really don't have much knowledge," however she immediately talked about "the activities you had us do with the tape. Then we could teach our students the stuff that directly affects them and how [economics] affected or was a product of the Delano Grape Strike and the economics behind it." This activity helped her because that was "the way I learn and in the process I'm confident enough to say, 'Okay I feel like I understand more things or vocabulary or the idea around this topic so I can go ahead and teach it to my students'" (interview, 7/13/17). So, not only was economic content presented in a way that was helpful, she saw the content and the pedagogy as directly applicable and relevant for her students. For Lizeth, the session exposed her to new concepts that expanded her purpose for teaching economics. According to her:

I hadn't really thought about before this summer the effect of inflation on the present and how things keep increasing in price yet the minimum wage doesn't have that much of a difference over time. And how that affects so many people right now, so many people who are just struggling to get by. (interview, 7/30/17)

The specific content of income inequality, and the temporal connection of the past to the present kickstarted a new line of thinking for the preservice teachers, allowing them to use economics as a tool for *social analysis* in the past and connect that information to the present. This led to a positive experience in the classroom where "I did enjoy teaching them that lesson about inflation and minimum wage because I was learning it with them as well. Then some of my students were using it in their presentations of learning projects" (interview, 7/30/17). Tori also highlighted the session as expanding her perception of the function of economics. When asked how her purpose for teaching economics might have changed after some time in the classroom, she replied:

It's changed a little bit in the sense that thinking of what to do with money [but] it also could be like looking at the rise and fall of incomes in neighborhoods. So instead of just the banking system, also money in general and how it affects populations. (interview, 7/20/17)

Upon follow up, this change was less a result of time in the classroom and more about "the PD review" (interview, 7/20/17). By directly weaving together new economic content with an expanded use of economics that fit their purpose, the professional development session held an outsized import in the expansion of these preservice teachers' understanding of the purpose of economics. It also allowed them to be more comfortable with the content while understanding society along a timeline extending into the past and the future.

## 4. Discussion

This study explored the question of how content knowledge and previous experience with economics influenced the way that preservice teachers understand the function of economics within social studies education. This exploration utilized the theoretical lens of *social analysis* as part of a critical pedagogical practice and *counter-hegemonic stance* as part of a transformative purpose for teaching. The data reveal two significant themes. One, preservice teachers' purpose for teaching social studies and the function of economics were aligned in an analysis of society that occurred in the present as they described the need to be aware of social forces, to understand the way social forces operate, and to conceptualize justice. Two, preservice teachers' purposes for social studies extended beyond the function of economics into the past, and informed active citizenship for future action, yet economics did not function in this way. The exploration of these themes led to the emergence of three common findings. First, economics can be a significant part of a social studies education practice that seeks to analyze society, understand the past, and take action for a better future. Second, limited familiarity and content knowledge inhibit a broader application of the function of economics into an exploration of the past and as a component of active citizenship. Third, social studies teacher education must purposefully integrate economics content into the exploration of the past and discussion of citizenship and action for justice in order to combat prevailing content knowledge issues in preservice teachers and to help them reconcile their purpose for teaching social studies through economics.

### 4.1. The Utility of Economics within a Transformative Social Studies Purpose

John Smyth [83] writes that "[a] truly critical pedagogy involves an examination of existing social relationships at three levels: that of history, of current practice (including its hierarchical bases and of the potential to transform arrangements in the future" (p. 21). The *social analysis* that teachers described as the purpose of social studies occurred on these three levels. However, when speaking about the function of economics within their purpose for teaching social studies they could only "expose these power-related dynamics that prop up the status quo, undermine social mobility, and produced discourses, and ideologies that justify such antidemocratic practices" [60] (p. 100) in the present. Yet this indicates that for a critically-minded social studies educator, the inclusion of economics has a vital function as part of a *counter-hegemonic stance* informed by a political clarity that recognizes the "sociopolitical and economic realities that shape lives and their capacity to transform such material and symbolic conditions" [37] (p. 98). Unfortunately, the discipline of economics provides limited support in extending economic analysis into the past [75–77], and rarely takes up social issues of race, class, and gender [84–87] that might be relevant to an active pursuit of justice as a citizen. Therefore, critically minded teachers, teacher educators, and preservice teachers should consider the temporal connections between their purposes for teaching and the way that economics functions within those purposes as part of a transformation based system of teacher preparation [88].

### 4.2. Confronting the Content Knowledge Gap and Its Impact on the Function of Economics

Social studies teachers and preservice social studies teachers have too often received too little exposure to economics content prior to their teacher preparation program and eventual teaching experience [3,12,14,56,57]. It should come as no surprise that this lack of familiarity can have a deleterious effect on student learning in economics [89,90]. King and Finley [91] describe the utility of Critical Race Theory in economics as "*the ability to understand and critique economic systems, recognize the inherent racism existent within the U.S. free market or capitalist economic system, and enact strategies that overcome obstacles presented by racist economic systems*" (emphasis in original, p. 203). This recognition of the potential of economics as *social analysis* tool as well as an important component of a *counter-hegemonic stance* with respect to race applies to other axes of oppression as well.

Economics can aid in a historical understandings of topics as diverse as gender inequality [92], heterosexism in the labor market [93], and the intersection of religion and wages [94]. It can also

enable students to conceptualize active citizenship practices that seek to transform society [95–97]. In this study, the lack of content knowledge and previous experience with economics inhibited the potential for preservice teachers utilize an economic lens in accordance with their purpose for teaching social studies. Their desire to analyze society in the past and present with the goal of utilizing active citizenship to transform the future was clearly stated, yet economics rarely broke the chronological bounds of the present as a factor in this purpose. Attending to the limited content knowledge that many preservice teachers demonstrate with respect to economics should consider this schism an opportunity to explicitly utilize economics to flesh out a critical social analysis of the past and as a vital component of active citizenship in the future.

### 4.3. Purposeful Integration as Part of a Counter-Hegemonic Stance

In order to enhance the content knowledge of preservice social studies teachers, efforts have been made to restructure teacher certification programs [98], to collaborate with faculty in other departments [99], and, with respect to economics, to offer content-specific methods courses [12,13]. These attempts to address the prevailing lack of content knowledge in economics are important to consider, but fundamental changes to teacher education practices may not always be feasible. Within the existing structures of social studies teacher education, it is important to consider ways to efficiently address limited content knowledge.

Teacher educators who intend to promote a humanizing social studies teacher education program that develops the "political and ideological clarity that will guide [preservice teachers] in denouncing discriminatory school and social conditions and practices" [37] (p. 119) must do so throughout their teacher education program [88]. This includes a focus on the way preservice teachers see the world and questions dominant assumptions [22], the practices they implement in the classroom [23,33], and the emphasis they put on praxis as the goal of seeing the world and humanizing classroom practices [100,101]. Therefore, social studies teacher educators must understand and support the transformative purpose of their preservice teachers, find economics content that fits with the content they will be teaching and model the implementation of critical pedagogy in methods courses. This multidimensional, integrated approach holds promise for improving content knowledge, pedagogical knowledge, and pedagogical content knowledge while supporting the development of a counter-hegemonic stance.

### 5. Conclusions

Preservice teachers in this study had very little prior exposure to economics, either in the dominant, neoclassical narrative, or a more critical version of economics. However, their conceptualization of the purpose of social studies focused on a critical use of social studies for *social analysis* in the past, present, and future through active citizenship. The function of economics within this budding emphasis on a *counter-hegemonic stance* was limited by the lack of familiarity with the discipline. While these preservice teachers make up a very small case in a very specific teacher preparation program, those social studies teachers and teacher educators who are concerned with critical pedagogy and transformative teaching can use their example to better conceptualize the role of economics. Supporting preservice teachers' critical vision of social studies practice means emphasizing the efficacy of economics as part of a transformative purpose, including the ways that economics can help understand injustice in the past and promote a more just future. It also means infusing economics in a purposeful way, with attention to modeling classroom practices while exposing preservice teachers to unfamiliar content. A counter-narrative in economics is possible, but only if preservice teachers can expand their understanding of economics within their transformative purpose and through their critical pedagogy.

**Funding:** This research received no external funding.

**Conflicts of Interest:** The author declares no conflict of interest beyond those addressed in Section 2.4-Researcher Positionality.

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
