# Peer review of "Economics (Only) Now: The Temporal Limitations of Economics as Part of a Critical Social Studies Pedagogy"

_education, doi:10.3390/educsci9010036_

Reviewer 1 Report

This manuscript makes a valuable contribution to the literature on critical pedagogy, social studies education and social studies teacher education. Below are a few recommendations for revision that I think will enhance the overall quality of the work.

Add connection between opening quotation and the "now" of economics.

Expand and revise theoretical framework. The author opens the section on the theoretical framework with reference to participants in the study; however, the participants have not been introduced yet and more importantly, this section should explain that concepts that guided the study. In the Theoretical Framework section, the concepts of social analysis and counter-hegemonic stance need to be introduced and explain independent of any reference to findings.

3. The author includes a sub-section titled Content Knowledge and Previous Experience with Economics in the Methodology section of the manuscript. This section should be moved to the Results and Discussion section. The author learned about the participants' experiences with economics through the interviews conducted with participants. I recommend this section become the first sub-section in the Results and Discussion portion of the manuscript. Also, it might be helpful to introduce each participant individually in the Participants and Setting sub-section.

4.  Provide more information about the 'variety of artifacts' that were also collected as part of the data set.  

5. I recommend re-working the results section to make the comparison between the differences between the ways in which the participants conceptualized economics and social studies the primary 'thread' throughout the section. I don't think the current organization of the results section clearly and succinctly presents the findings. 

Author Response

Response to Reviewer 1 Comments:

1.     Add connection between opening quotation and the "now" of economics

a.     Response: As other changes to the manuscript necessitated more writing, I chose to excise this quotation (as well as the one prior to the conclusion) in the interest of space.

2.     Expand and revise theoretical framework. The author opens the section on the theoretical framework with reference to participants in the study; however, the participants have not been introduced yet and more importantly, this section should explain that concepts that guided the study. In the Theoretical Framework section, the concepts of social analysis and counter-hegemonic stance need to be introduced and explain independent of any reference to findings.

a.     Response: These opening sentences were removed from the section, and the information therein was included in the relocated sub-section on content knowledge (according to suggestion 3). Further exploration of both social analysis and counter-hegemonic stance were added to the theoretical framework section.

3.     The author includes a sub-section titled Content Knowledge and Previous Experience with Economics in the Methodology section of the manuscript. This section should be moved to the Results and Discussion section. The author learned about the participants' experiences with economics through the interviews conducted with participants. I recommend this section become the first sub-section in the Results and Discussion portion of the manuscript. Also, it might be helpful to introduce each participant individually in the Participants and Setting sub-section.

a.     Response: Section moved and detail added for each participant in the Participants and Setting sub-section.

4.     Provide more information about the 'variety of artifacts' that were also collected as part of the data set.  

a.     Response: It was perhaps unclear that the entire paragraph that followed was describing those artifacts. I rewrote the transition to better illustrate that the following elements (reflections, coursework, materials from professional development) were the artifacts described and expanded some description of those elements.

5.     I recommend re-working the results section to make the comparison between the differences between the ways in which the participants conceptualized economics and social studies the primary 'thread' throughout the section. I don't think the current organization of the results section clearly and succinctly presents the findings. 

a.     Response: I am most thankful for this suggestion, as I have struggled to find the appropriate way to frame the results. I have separated the two descriptions of purposes to make that the primary thread and believe it has had the intended effect of adding clarity and succinctness to the findings.

Reviewer 2 Report

The concept is definitely interesting and important.  Economics is a critical part of social studies, but is notoriously poorly taught (and frequently focused only on macroeconomics).  As you mention in the paper, teachers tend to be woefully under prepared to teach economics.

That said, the paper seems to present a derisive attitude towards "neoclassical economics" that colors the analysis.  It is not entirely clear why this is the case.  One of the common problems with those underprepared in economics is the inability to distinguish positive analysis (e.g. the optimizing behavior and utility theory that underlie all human choices) from normative analysis (e.g. policy issues that require judgements based on what should be).

Teachers need to be able to instruct students on the positive economic way of thinking, so that they can then have intelligent discussions about normative issues (and perhaps then consider orthodox versus heterodox economic models).

Methodologically, this paper is a very small case study of a very small number of students in a very specific environment.  Any conclusions apply only to this very specific situation.

The language used is needlessly flowery.

very end of line 53, 'economic' should be 'economics'.

line 56 and 59, also 'economic' should be 'economics'.

line 107 what 'previous chapter'?

Author Response

Response to Reviewer 2 Comments:

1.     That said, the paper seems to present a derisive attitude towards "neoclassical economics" that colors the analysis.  It is not entirely clear why this is the case.  One of the common problems with those underprepared in economics is the inability to distinguish positive analysis (e.g. the optimizing behavior and utility theory that underlie all human choices) from normative analysis (e.g. policy issues that require judgements based on what should be).  Teachers need to be able to instruct students on the positive economic way of thinking, so that they can then have intelligent discussions about normative issues (and perhaps then consider orthodox versus heterodox economic models).

a.     Response: While it is correct to say that this paper specifically addresses neoclassical economics, I am not entirely sure that it is derisive, nor that it colors the analysis. I searched for each mention of ‘neoclassical economics’ in the paper to confirm, and I think what I was trying to do is make explicit the distinction between orthodox and heterodox economics that you describe (something that is unfortunately rare in social studies education literature). Specifically, I point out that neoclassical economics has become economics if one looks at the textbooks and standards that preservice social studies teachers are likely to come in contact with.

While a number of theorists (in economics literature, again, rarely social studies education literature) point out the deleterious consequences of this orthodoxy (c.f. Marianne Ferber, Julie Nelson, or countless others who come from a feminist lens; the work of Rethinking Economics and their handbook and publications; and general critiques from heterodox and pluralist perspectives including Silja Graupe, Jacek Brant, Jakob Kapeller, Steve Keen, Frederic Lee and others) this paper does not do that. It simply reminds readers at every turn that when we talk about economic content knowledge among preservice teachers, we are usually talking about neoclassical economic content knowledge, and by NOT making that explicit we subsume any possibility for intelligent discussions about the relative merits of a variety of economic paradigms and models. I think you are correct in your appraisal that we struggle to teach the difference between positive and normative analyses, but we also should remind students that not all economists are convinced that optimizing behavior and utility theory underlie all choices.

2.     Methodologically, this paper is a very small case study of a very small number of students in a very specific environment.  Any conclusions apply only to this very specific situation.

a.     Response: Changed subsection on researcher positionality to include limitations of the study and added a short paragraph to make explicit the nature of conclusions to follow. Additionally, in conclusion section, I referred to this limitation when making final points about the implications of the study.

3.     The language used is needlessly flowery.

a.     Response: I think this refers to some of the subheadings in the results section that were lyrics from Marvin Gaye’s seminal album What’s Going On (used as the title for the section). Given changes to the organization of the results section suggested by another reviewer, these headings were renamed as part of the reorganization and to avoid the unnecessary (and unhelpful) allusion that was not very well connected. I am happy to make any other changes to the body of the text, but would need specific examples to address.

4.     very end of line 53, 'economic' should be 'economics'.

a.     Response: Changed.

5.     line 56 and 59, also 'economic' should be 'economics'.

a.     Response: Changed.

6.     line 107 what 'previous chapter'?

a.     Response: Removed.

Round  2

Reviewer 2 Report

None